# The European Educational Policy and Early School Leaving: A Conceptual Analysis from the Perspective of the Leaving Subject

Laura Guerrero Puerta [1,2]

1   Department of Didactics, School Organization and Special Didactics, National University of Distance Education (UNED), 18071 Madrid, Spain; laura.guerrero.puerta@edu.uned.es
2   Reseach Group HUM-308, University of Granada, 52005 Granada, Spain

**Abstract:** This article presents a conceptual analysis of the European educational policy concerning the phenomenon of early school leaving (ESL). It addresses the literature on ESL, emphasizing the importance of studying policies from the perspective of the constructions made of the leaving subject. The concept of lifelong learning is examined, along with its relevance in shaping the subject who leaves within European policies. Additionally, the presence of "double gestures" in educational policies is explored, where, while promoting inclusion, they simultaneously produce exclusion of certain individuals. The influence of the neoliberal approach on the conception of ESL is discussed, and the need to consider alternative approaches to avoid homogenizing the group of young individuals who leave school prematurely is proposed.

**Keywords:** early school leaving; knowledge society; lifelong learning; philosophy of education; youth

## 1. Introduction

The phenomenon of early school leaving (ESL) has been a topic of attention in both academic literature and educational policies in Europe. However, there is a recognized need to analyze this issue from a more conceptual perspective to understand the constructions made of the leaving subject within the framework of educational policy. In this context, this article aims to address early school leaving from a philosophical perspective, exploring the conceptual implications behind European educational policies and their relation to the construction of the leaver. Through critical and reflective analysis, we seek to understand how the notion of the "leaver" is shaped as an individual who does not conform to the ideals of the knowledge society and lifelong learning.

Firstly, we will review the literature on early school leaving, emphasizing the need to question the numerical reductionism present in European discourse and the promotion of neoliberal values. From there, we will delve into the concept of lifelong learning and its relevance in shaping the leaver within educational policies. We will explore feminist and Marxist criticisms of this conceptualization, highlighting how barriers and inequalities faced by certain groups in the learning process are ignored.

Subsequently, we will examine the notion of "double gestures" in educational policies, revealing how, despite promoting inclusion, certain subjects are also excluded. From a philosophical perspective, we will reflect on the ethical and political implications of this ambivalence and its impact on the leaver.

Finally, we will analyze the influence of the neoliberal approach in the conception of early school leaving, questioning individual responsibility and the exclusion of other perspectives to address this issue. We will propose a more critical and reflective outlook, considering the complexities and diversities of the leavers, aiming for a more inclusive and equitable education.

In this way, this article seeks to contribute to the philosophical debate on European educational policy and early school leaving, inviting deep reflection on how the leaver is

constructed within the context of the knowledge society and lifelong learning. Through a philosophical lens, we aim to enrich the analysis and promote a fair and respectful education that acknowledges the diversity of individuals within our society.

## 2. Historical Analysis of Early School Leaving (ESL)

If we take a historical view of the educational environment, we can observe that, recurrently and regardless of the specific context, a portion of the student population attending school does not complete their education and leaves the educational system to enter the labor market. Some authors, such as Escudero et al. [1], even argue that early school leaving is as old as schooling itself. While this claim may hold some truth, it is also evident that, relatively recently in the social and political imaginary, leaving school without achieving a certain level of educational certification has acquired a new significance, emerging as one of the main educational issues whose effects not only concern the educational realm, but also impact economic and social domains.

Therefore, we can say that premature departure from the educational system has been problematized, and the genesis of this as a problem is relatively recent. This problematization, which has been barely explored in Spain [2], holds great importance, as it will have a clear effect on the life course of the young people who leave school. They may assimilate existing discourses into their own identities through a process widely explored by Bourdieu [3,4], involving the incorporation of social structures around early school leaving in the form of dispositional structures (habitus) and objective chances such as expectations and anticipations, which will shape their life trajectories.

The emergence of a social problem does not occur spontaneously; no social problem emerges or institutionalizes without a series of transformations in the objective conditions and structures of a society [2]. Typically, it is the result of a combination of various objective transformations, collective enunciation, and mobilization, which promote the development of categories and identities that may or may not succeed in gaining recognition and legitimation as a problem, as well as a process of institutionalization [5]. As Foucault points out, these transformations, encompassing social, cultural, demographic, and economic aspects, play a crucial role as instigating factors. Their impact often begins long before the problematization takes root in the collective consciousness. In essence, for a particular field of action or behavior to become a subject of contemplation, a multitude of factors must render it uncertain, strip away its familiarity, or generate various challenges around it [6].

Thus, for early school leaving to have attained the status of a problem in most European states, a series of transformations beyond the educational sphere have been necessary to enable it. The development of this process can be explained through the proposed phases, which are based on the developmental process proposed by Rujas [7]. This article, however, includes an additional third stage, following the ideas of authors like Leathwood [8], Blackmore [9] or Gillies and Mifsud [10], who propose that the effects of the Lisbon Strategy and its efforts to achieve a knowledge-based economy have transformed the understanding of early school leaving from a national problem to a communal problem. This shift demands greater consideration of this process, as it significantly alters the definitions and comprehension surrounding this issue, and has a substantial impact on the life trajectories of those involved.

### 2.1. Initial Stage: Universalization of Secondary Education

There is a certain consensus [7,11,12], in indicating the universalization of education as one of the primary conditions that have positioned early school leaving as an international concern, associated with the establishment of new "specific age norms". This is not a coincidence, as until the Second World War, child labor was prevalent in European states, and a significant portion of the child population, mostly from working-class backgrounds, either remained unenrolled in school or attended school for a brief period before leaving to enter the workforce to support their families. However, the Great Depression following the Second World War significantly changed the production model, leading to children

and adolescents distancing themselves from the labor market. This forced a delay in the age of workforce entry, after which attending school and completing secondary education gradually became a social norm (i.e., something normal or customary in society) [11].

In this sense, Dorn [12], an American historian of educational policies known for his studies on the origin of the theme, points out that the beginning of this process of problematization of early school leaving can be traced back to around the 1960s, a decade after the development of the first educational systems with a relatively widespread secondary education [13].

While scholars such as Isambert-Jamati [14], Ravon [15], Rujas [7], or Morel [16] support this timeframe, they also note differences among European states that attributed to varying rates of expansion in formal education. They highlight states like the United Kingdom, Germany, and France, whose educational systems were pioneering in Europe thanks to early industrialization and intellectual and religious movements that regulated schooling for the population at an earlier stage. In contrast, other states, such as Spain, had to wait several decades for a more or less universalized education system to emerge, due to greater political instability and slower industrial development [17].

Despite these slight variations in temporal development, a consensus exists that the beginning of this problematization is materialized when secondary education became associated with primary education as a common and standard practice. This is evident in the early writings of teachers and researchers, who began to express concern about children who "were unable to adapt to the school environment," adopting an individual deficit perspective [7].

During this phase, early school leaving became associated with gender stereotypes, perhaps stemming from a tradition, present since Rousseau's "Emile," of segregated education, where the educational and career paths of boys and girls were treated differently. Coupled with a moral panic around adolescents and delinquency, which was associated with males, much of the attention given to this emerging issue focused on boys who dropped out, under the belief that they were more dangerous.

### 2.2. Second Stage: Genesis and Crystallization of Early School Leaving as a National Problem

Between the 1970s and 1980s, and following the initial enunciation phase, a second stage in the development of early school leaving as a problem took place. Described by Rujas [7] as the Genesis stage, using the Foucauldian concept, this stage occurred thanks to the revitalization of institutions focused on childhood and youth problems, changes and restructurings in state political and educational systems, as well as variations in class systems, resulting in an overall increase in middle-class participation in schooling.

During this time, concepts associated with early school leaving, such as academic failure and school difficulties, began to appear in national spaces. Gradually, they became associated with obtaining specific qualifications or credentials. The academic landscape witnessed the emergence of the first discourses around human capital and economic growth, which argued for the existence of a factor of production dependent on quality, productivity, and educational level. Consequently, high rates of early school leaving began to be perceived as a "waste" of this potential, and political debates emerged about the ineffectiveness of school systems [2,7].

As a result, national educational systems began to establish their own measures to combat early school leaving, and new educational paths were developed for students who were not "qualified" to receive secondary education certification and continue on to university. Additionally, around the 1980s, indicators measuring the school failure rate started to emerge, which led to a certain process of "labeling" those who did not complete this stage. As a consequence, certain understandings of the leaving subject became evident in political discourse and social imagination. This was the moment when early school leaving, according to Rujas [7], crystallized as a public problem.

*2.3. Early School Leaving (ESL) from a National Problem to a Community Problem*

In the European context, from the 1990s onwards, early school leaving has gradually become a political priority. Some authors [10,18,19]; highlight that in this process, discourses centered around justice and social cohesion have been intertwined with overtly economic discourses, with the latter becoming increasingly dominant. As a result, early school leaving has been problematized from a utilitarian standpoint, focusing more on its effects on economic growth and employment. In this perspective, those who leave school are classified as a "risk" due to lacking the necessary skills for employability, leading to a tendency to "blame the victims". This process has been identified in the literature through two phases. The first phase (1) was initiated by the Lisbon Strategy, where the shift towards a knowledge-based society and the move towards governance based on the open coordination method played a key role. The second phase (2) was initiated with the Europe 2010 Strategy and continued through the Europe 2020 Strategy, where, under the influence of the 2008 economic crisis, there was an even clearer turn towards the pursuit of youth employability from a more individualistic perspective, placing a greater emphasis on early school leaving. These stages are explored below.

2.3.1. The Seed: The Lisbon Strategy and the "Education and Training 2010"

During the year 2000, as part of the Lisbon Council Strategy, the objective of making Europe "the most competitive and dynamic knowledge-based economy in the world, capable of sustainable economic growth with better jobs and greater social cohesion" was established [20]. Until this point, the first phase of European coordination (from the Maastricht Treaty in 1992 to the Lisbon Strategy) had been cautious and hesitant, especially in the field of education, although clear about the need for common educational strategies and new ways of understanding education. The Lisbon Strategy is identified by authors like Novoa [21] as a "point of no return" in terms of a new way of approaching the relationship between education and the labor market.

The importance of this event can be summarized in two decisions that were reached, and which have had a highly influential impact on the development of subsequent European strategies. These decisions specifically relate to the reconstruction of the social perception of early school leaving. These decisions are as follows:

(A)　Establishment of the knowledge-based society

The knowledge-based society/economy appears as an open and evolving concept, which must be defined by the community and embraced as a co-constructed and identity-building project with and for European citizens. This idea is clearly represented in the following excerpt:

> "Europe must find its own way to build a society and economy based on innovation and knowledge. The European path must open up opportunities for access to knowledge, value cultural diversity to a fair extent, and use this transition to better shape a specific European identity and to identify citizens more with a European project that they will define themselves" [22][1].

Although this concept is presented as open and interchangeable, the discourse of the knowledge-based society/economy in Europe is linked to a process of seeking a European identity and legitimizing European values that citizens are exhorted to construct. Additionally, this new society is presented as a result of the need to modify a social and economic model that had become obsolete in the face of new labor market conditions and existing societies that required technological renewal and an emphasis on innovation [23]:

> "The Union has set a new strategic objective for the next decade: to become the most competitive and dynamic knowledge-based economy in the world based on knowledge, capable of sustainable economic growth with more and better jobs and greater social cohesion. Achieving this objective requires a global strategy aimed at: preparing the transition to a knowledge-based economy and

society through better policies for the information society and R&D, as well as intensifying the process of structural reform for competitiveness and innovation and completing the internal market; modernizing the European social model, investing in people and combating social exclusion, sustaining sound economic prospects and favorable growth prospects through the application of an adequate combination of macroeconomic policies. . ." [20][2].

One of the characteristics of this new knowledge-based society at this historical moment is that it proposes a social model that incorporates discourses about seeking social cohesion, but these are constructed in economic terms and within a backdrop of crisis. This highlights the urgent need to build a new strategy to address the disappearance of Fordist models [23]:

> "As long as this lack of adaptation to the new paradigm continues, there will be a deficit of economic growth and a greater risk of unemployment and social exclusion. We have to increase the pace of technological change but also of institutional reform... An economic and social strategy to renew the foundations of growth in Europe must combine macroeconomic policies, economic reform, structural policies, active employment policies, and the modernization of social protection" [22] (p. 5[3]).

During this period, two documents were published that introduced a key relationship for understanding the subsequent evolution of European policy in terms of employment and education: the Memorandum on Lifelong Learning [24], and the European Report on the Quality of School Education [25]. The concept of "lifelong learning" emerged, reconfiguring "all learning as a seamless continuum 'from cradle to grave'[24]. Lifelong learning was presented as a key factor in developing "employability," the basis for Active Employment Policies, and was conceived as a means to promote quality employment [21].

(B)　Open Coordination Method

The second decision that would change the course of European policy was the establishment of the Open Coordination Method, defined as "a form of 'soft' legislation. A form of intergovernmental policy-making that does not lead to binding EU legislation and does not require EU countries to introduce or modify their laws" [26]. This method was based on "establishing common objectives to be achieved (adopted by the Council)," designing "measurement instruments (statistics, indicators, guidelines)," and a "comparative evaluation, i.e., comparing the results of EU countries and exchanging best practices (supervised by the Commission)" [26].

This marked the beginning of designing indicators and benchmarks, including early school leaving and other concepts related to employment and education. These indicators were developed with the aim of enabling adhering states to "learn from each other, share successes and failures, and collectively use education to move towards the new millennium" [21,27].

Although early school leaving had not yet gained the prominence it would later acquire during this period, we can consider this period as germinal for several reasons: (1) The introduction of the open coordination model marked the beginning of a new governance model heavily conditioned by the use of indicators and benchmarks, which would become key in policy design processes in different states and started to focus on education. (2) It changed the rules regarding the learning process, expanding the scope of education from a limited period to a lifelong process. (3) Lifelong learning began to be associated as a key factor for success in the labor market, albeit from a humanistic and structural perspective, emphasizing the creation of quality jobs. (4) Lastly, as part of the project of the knowledge-based society, the idea emerged that citizens play a crucial role in its construction and success, promoting a new model of citizenship with greater responsibility.

2.3.2. From the Lisbon Strategy to the Present

Shortly after the Lisbon Strategy, in 2002, the "Education and Training 2010 (ET2010)" strategy was presented, aiming to specify the general strategic objectives that had been proposed in 2001 through the "concrete future objectives of the education and training system" report. ET2010 outlined five common strategic objectives that member states should work on regarding education: (1) ensuring that European education and training gain global recognition for their quality and relevance; (2) guaranteeing that education and training systems have mechanisms for mobility; (3) ensuring that qualifications held by citizens are valid throughout the European Union; (4) guaranteeing access to lifelong learning for all citizens; and (5) promoting cooperation and mobility with other regions of the world. The importance of exchanging good practices through the open coordination method for achieving these objectives was also emphasized. A year later, with the publication of the strategic monitoring indicators of ET2010, early school leaving began to gain more prominence, as these indicators included for the first time a reduction in early school leaving to a maximum of 10% [28].

Beyond the role attributed to early school leaving, Novoa [21] presents interesting reflections on this period, which he sees as a point of no return regarding the new European governance model based on indicators and "soft law," making it difficult for different states to not integrate into this proposal. Additionally, the concept of employability is reinvented as a way to link employment and education, interpreting unemployment as a problem of "uneducated" individuals. Thus, if this concept was originally linked to the pursuit of job quality, it is now presented in a way that shifts responsibility from the political system to citizens, who must constantly update their knowledge to achieve "employability". This, in turn, redefines the concept of lifelong learning as a response to this individualization of employability. As we will see later, this will have significant implications for the concept of early school leaving.

In 2009, as many of the objectives proposed in ET2010 were not achieved, a new program called "Education and Training 2020" (ET2020) was accepted to support EU member states in developing their educational systems [29]. It consisted of four common strategic objectives: (1) Improving the quality and efficiency of education and training; (2) promoting equity, social cohesion, and active citizenship; (3) making lifelong learning and mobility a reality; and (4) enhancing creativity and innovation, including entrepreneurship, at all levels of education and training. The research on early school leaving was framed in relation to these four strategic objectives, but more directly connected to the third objective, which stated that "education and training systems should strive to ensure that all learners, including those from disadvantaged backgrounds, complete their education". By making early school leaving a priority area for EU cooperation in the period 2010–2020, a clear mandate was given to the European Commission, EU member states, and all other relevant stakeholders to collaborate closely, resulting in various activities and the acceptance of numerous indicative (non-binding) documents providing general and more concrete guidelines for jointly achieving the EU's early school leaving goal.

Novoa [21] points out that through ET2020, the impact of the crisis on the European continent can be observed. This, combined with the growth of emerging countries such as China, Brazil, or India and the inability to achieve proposed objectives in previous stages, led to a loss of ambition to become "the most competitive economy in the world" and, in general, a strategy designed more for continuity rather than a radical breakthrough or great innovation. However, he also notes that within this crisis context and as a strategy to combat youth unemployment, the strategy redefines education, taking a more pronounced shift towards neoliberal positions by contextualizing educational policies based on promoting the skills needed for the labor market or entrepreneurship.

In November 2012, a remarkable document titled "Rethinking Education: Investing in skills for better socio-economic outcomes" [30] was published. This document is considered by Ross and Lethwood [18,19] as one of the most remarkable points in the European agenda regarding combating early school leaving. For the first time, a connection was established

between the interest and investment being made to address early school leaving and the economic rationality underlying this strategy. This connection is represented in the following excerpts, which explicitly state that the most urgent function of education is "to respond to the needs of the economy and focus on solutions that can reduce the growing youth unemployment (p.2)". It further mentions the need to "provide the skills necessary for employment, increasing the efficiency and inclusiveness of education," and "improving the performance of students who belong to groups at high risk of early school leaving and have low basic skills". This is identified by the authors as the most recent and surprising exposition of the neoliberal agenda for education, explicitly linking the justification for investing in education to economic growth from the perspective of education and training policies. The following section will explore the implications of this approach.

Fast forward to the "strategic framework for European cooperation in education and training towards the European Education Area and beyond (2021–2030)" [31], we find an approach that, while recognizing the impact of the COVID-19 pandemic on education, falls short in addressing the structural issues related to early school leaving and educational inequalities. It emphasizes the European Education Area and lifelong learning but lacks a critical examination of their neoliberal dimensions, which often places the burden of employability on individuals and perpetuates inequality. Furthermore, the framework's mention of the European Pillar of Social Rights is a positive step, but it requires concrete strategies to ensure equal opportunities and social fairness in education, especially for marginalized groups. The emphasis on the green and digital transitions is crucial, but it needs a more explicit commitment to equitable implementation. In essence, the framework must navigate the complex terrain of educational policy evolution in Europe, recognizing the historical roots of economic rationality in education while striving for a more socially just and inclusive future.

### 3. European Educational Policy: Analyzing the Construction of the Leaving Subject

The literature on early school leaving has begun to call for a greater emphasis on the study of European policies from the scrutiny of the constructions made about the subject who abandons. For example, Bansel [32] rejects the numerical reductionism used in European discourse, arguing that it promotes acceptance of neoliberal values. He calls for an analysis that explores the type of subject that is "embodied" in policies through the relationships introduced in them. Gilles and Mifsud [10], on the other hand, argue for the urgency of an interpretation of how the "being" who abandons is constituted, and what possibilities of manifestation are attributed to this subject. They ask that, through the exploration of these underlying discourses in policies, along with the observation of the interrelationships between the different concepts present in them, specific life models associated with these individuals be brought to light, trying to discern between the multiple and the singular; the political body and the embodied subject; and the human and the non-human, in the face of a political narrative that, according to their argument, reduces the subject of these policies to a simplified and quantifiable object, causing the student who abandons to be manufactured within and through the discourse.

### 3.1. The Concept of Lifelong Learning and Its Relevance to Understanding the Construction of the Subject Who Abandons

The concept of lifelong learning, in its more closely linked conception to employability from an individualistic standpoint, positions itself as one of those key interrelations for studying how the subject who abandons is constructed in European policies. As Blackmore [9] indicates, political discourses that implicitly or explicitly take into account the conceptualization of lifelong learning as a key aspect of the knowledge society are promoting a particular way of learning to be, do, work, and learn based on the accumulation of credentials and competitiveness. This may lead us to think that certain groups will be excluded.

### 3.1.1. Critique of the Concept of Lifelong Learning from Feminist Studies

Feminist studies have explored how the concept of lifelong learning can be exclusionary. Authors such as Alheit [33], Leathwood [8], and Blackmore [34] highlight that these discourses ignore possible barriers to participation, social positioning, and internalized narratives that certain groups may face in the learning process. For example, the responsibility of caring for children and the household can be a significant barrier for some women to access and participate in lifelong learning, especially in the absence of structural support that reduces the burden of these tasks that still disproportionately fall on women. Additionally, certain stereotypes and values associated with women's participation in lifelong learning can further limit their access and engagement. This critique suggests that the logic of lifelong learning, despite its explicit reference to the pursuit of social justice, minimizes or even overlooks certain inequalities, perpetuating the status quo for certain groups [8].

This critical examination of the concept of lifelong learning in the context of feminist scholarship raises fundamental questions about the meaning of social justice within education. Social justice in education encompasses the principles of equity, fairness, and inclusion [9]. It calls for the removal of systemic barriers that hinder individuals from marginalized groups, such as women, from accessing educational opportunities on an equal footing with others [35–37].

At its core, social justice in education demands the recognition and rectification of disparities that may arise from societal norms, stereotypes, and structural inequalities. It challenges the status quo by advocating for policies and practices that empower individuals to overcome these barriers. In the case of lifelong learning, a socially just approach would involve recognizing the disproportionate caregiving responsibilities that often fall on women and taking proactive steps to provide the necessary support, such as affordable childcare or flexible scheduling, to enable their full participation in educational pursuits [38].

Furthermore, social justice in education requires a commitment to addressing the implicit biases and societal attitudes that hinder the engagement of marginalized groups. By challenging stereotypes and values that may limit women's participation in lifelong learning, a socially just educational system aims to create an inclusive and supportive environment where all individuals have the opportunity to develop their full [38].

In summary, social justice in education, as illuminated by feminist critique, emphasizes the importance of dismantling systemic inequalities, addressing structural barriers, and challenging discriminatory norms and values. It seeks to create an educational landscape where everyone, regardless of their gender or other identity factors, can engage in lifelong learning and pursue personal and societal growth on an equal and equitable basis [38].

### 3.1.2. Critique of the Concept of Lifelong Learning from Marxist Studies

From a Marxist perspective, authors like Morley [39] criticize the concept of lifelong learning in the European context as a "seductive discourse" focused more on wealth creation than redistribution. By ignoring participation barriers and social positioning, it promotes "psychological narratives" centered on self-confidence, effort, and individual freedom, which can lead to "internalized oppression". This discourse aligns with neoliberal ideologies and introduces new forms of coercion and social exclusion, as it discourages structural attributions for the difficulties faced by certain social groups [40–42].

This relationship with neoliberalism and the new power dynamics it establishes has been widely explored in the literature. Philosopher Byung-Chul Han [43,44], for example, argues that under neoliberalism, the worker becomes an entrepreneur, with each individual exploiting themselves in their own "company". He describes how this neoliberal logic replaces the why with the what, quantifying reality in search of data, which leads to the expulsion of the spirit of knowledge. This process introduces new forms of domination, hidden from the individual, and fosters a conformity that eliminates the otherness, causing individuals who fail in the "neoliberal society of performance" to internalize blame and shame rather than questioning the system [43,44].

Given these perspectives, it becomes evident that the concept of lifelong learning in its current form tends to overlook the different costs and benefits of participation, and the various opportunities individuals have for engaging in lifelong learning. Education and learning are presented as neutral tools that promote social cohesion without considering how factors like gender, class, race, and others interact with the learning process [8]. This alignment with neoliberal discourses narrows the conceptualization of the phenomenon and excludes alternative approaches, homogenizing the group of early school leavers [10].

*3.2. From Inclusion to Exclusion: Double Gestures in European Policy and the Subjects Who Leave as Their Protagonist*

The concept of lifelong learning has promised a "virtuous circle of learning" in which learners receive returns on their investment in the form of money, prestige, and power, requiring them to continue learning. This notion is linked to the Matthew effect, exemplified in these discourses: "For to everyone who has, more will be given, and he will have an abundance. But from the one who has not, even what he has will be taken away". In this context, those who successfully participate in lifelong learning experience inclusion, while those who do not participate are excluded from obtaining returns. Thus, lifelong learning creates a circle of inclusion for participants and excludes those who do not engage in lifelong learning.

Thomas Popkewitz [45] explores the existence of "double gestures" in educational policies, where certain categories, like early school leavers, recognize a situation with the aim of promoting inclusion but simultaneously materialize division. These policies construct desirable and "natural" characteristics that define the ideal citizen in a given society while pathologizing those who do not fit these criteria, creating an "other," an outsider, based on characteristics contextualized by educational contexts [45–47].

Popkewitz's [45–47] analysis demonstrates how lifelong learning is an example of a double gesture in contemporary education policies. Lifelong learning discourse promotes inclusion, but at the same time, it can lead to exclusion. It presents an ideal learner as one who can solve problems and adapt to constant changes with agency and empowerment, participating actively in lifelong learning. Those who do not fit this ideal become the "other," the subject at risk and the one who jeopardizes community growth and cohesion. In this context, early school leavers can be perceived as embodying the "other," the subject who deviates from the lifelong learner ideal [10].

The construction of early school leaving in European policies has been intertwined with concepts such as employability and lifelong learning, understood in the context of developing the knowledge society. As a result, early school leaving is often framed as a problem hindering employability, or as the antagonist of the lifelong learner. This discourse may appear "logical" or "unquestionable," naturalizing the link between education and employment, but it is identified by some authors as a key factor in understanding the type of subject that falls under the definition of an early school leaver.

However, this way of understanding early school leaving and the subject constructed through it is rooted in a neoliberal framework that emphasizes individual responsibility during education-to-work transitions, obscuring the importance of government structures and promoting an individualization of responsibility. This narrow perspective excludes other approaches and homogenizes the group of early school leavers [10]. Further research is needed to understand how this impacts the life course of early school leavers, and whether it affects the process of early school leaving.

## 4. Critical Synthesis of Literature: Understanding the Multifaceted Nature of Early School Leaving (ESL)

In delving deeper into a critical synthesis of literature on early school leaving (ESL), we uncover a rich philosophical terrain that challenges our understanding of education, society, and individual agency. By examining the historical analysis and the various perspectives on ESL, we encounter fundamental questions about human nature, social structures, and the role of education in shaping life trajectories.

At its core, the problematization of ESL forces us to confront the dialectical relationship between the individual and society. As we trace the evolution of ESL from its historical roots to its contemporary manifestations, we discern how societal, economic, and political forces interact to construct the subject of who leaves school early. Here, the writings of Michel Foucault [6] become particularly illuminating, as he posits that social problems do not emerge spontaneously but are the result of intricate interplays of power and discourse.

Foucault's [6] notion of the "problematization" process sheds light on how early school leaving has come to be perceived as a critical issue. This process involves a series of transformations in objective conditions and structures of society, culminating in the emergence of discourses that define and categorize certain individuals as "early school leavers". These discourses are imbued with power, shaping the subject's identity and self-perception. The subject who leaves school early is not simply an individual making a choice; rather, they are positioned within a web of social norms, expectations, and systemic constraints.

Moreover, the construction of early school leaving as a problem underscores the power dynamics inherent in educational policies. The concept of lifelong learning, as a cornerstone of contemporary educational discourse, reveals the hegemonic neoliberal ideology that privileges economic growth and individual employability. This paradigmatic shift in education emphasizes the production of "human capital" and quantifiable outcomes, neglecting the broader humanistic goals of education and the cultivation of critical thinking and citizenship.

Drawing on the ideas of feminist and Marxist scholars, we are reminded of the inherent exclusions embedded within the concept of lifelong learning. Gender, class, and race intersect with educational access and participation, revealing that not all individuals have equal opportunities to engage in lifelong learning. The neoliberal narrative often masks these inequalities, attributing success or failure to individual merits and efforts, thus perpetuating a sense of internalized oppression.

As the philosopher Byung-Chul Han [43,44] noted, it is of a great concern the impact of neoliberal logic on subjectivity. The emphasis on self-exploitation and self-optimization under neoliberalism has profound psychological implications. Early school leavers, labeled as "at risk" individuals lacking employable skills, may internalize feelings of inadequacy and shame, further reinforcing the neoliberal narrative of blame on the individual.

In this way, from a critical theory perspective, the problematization of early school leaving highlights the role of power and ideology in shaping educational policies. The dominant discourse surrounding ESL may reinforce existing power structures and maintain social hierarchies. This raises questions about whose voices are included or excluded in shaping educational policies, and how marginalized groups may be disproportionately affected by such policies.

To achieve a more holistic understanding of early school leaving, we must engage in philosophical reflections that transcend quantitative data and policy analysis. We need to explore the underlying assumptions and values that inform educational systems and policies. By critically examining the historical context, the construction of discourses, and the implications for individual subjectivity, we can uncover the philosophical foundations that underpin the ESL discourse.

Ultimately, a philosophical analysis of early school leaving encourages us to reflect on the broader purpose of education and the societal responsibilities involved. It challenges us to consider how education can foster not only economic productivity but also human flourishing, social justice, and the cultivation of a critically engaged and empathetic citizenry. By embracing a philosophical perspective, we can transcend the limitations of reductionist approaches and work towards a more profound and transformative understanding of early school leaving and its significance for individuals and societies alike.

In light of this critical synthesis, we must consider education's broader purpose in society. Rather than reducing education to a mere tool for economic growth and competitiveness, we must reevaluate its role as a catalyst for human flourishing, social cohesion,

and critical engagement with the world. An alternative philosophical perspective emphasizes education as a transformative force that nurtures agency, empathy, and the ability to navigate complex societal challenges.

## 5. Conclusions

In conclusion, the critical synthesis of literature reveals that early school leaving is not merely an educational issue, but a complex social phenomenon influenced by historical, economic, and political transformations. The construction of the subject who leaves school early is heavily influenced by neoliberal discourses, which overlook structural barriers and reinforce exclusion. To offer a more holistic understanding of ESL, it is imperative to consider the interplay of societal, economic, and political factors that shape individuals' life trajectories and challenge the dominant neoliberal perspective. This nuanced and reflective analysis can contribute to the development of more inclusive and effective policies to address the multifaceted nature of the ESL problem.

The analysis of European educational policy through the lens of the leaving subject provides valuable insights into the complexities and implications of the discourses surrounding lifelong learning, employability, and social justice. By examining the representations of the leaving subject within these policies, it becomes evident that a narrow focus on numerical reductionism and neoliberal values can lead to the exclusion of certain social groups and perpetuate inequalities.

The construction of the leaving subject as an "other" who deviates from the ideal lifelong learner, highlights the need for a critical reevaluation of policy approaches. While European policies may appear inclusive in their aim to address school leaving, they also contribute to division and categorization, obscuring the structural barriers and complexities faced by certain social groups.

In addressing this issue, European policymakers must move beyond a one-size-fits-all approach and consider the interconnectedness of lifelong learning, employability, and social justice. By acknowledging the various barriers to participation and social positioning, policies can be crafted to promote more equitable and inclusive learning opportunities for all individuals, regardless of their backgrounds or circumstances.

Ultimately, a comprehensive understanding of the leaving subject and the underlying discourses within European policies is crucial for fostering a more just and inclusive education system. By challenging the neoliberal emphasis on individual responsibility and recognizing the structural factors influencing leaving decisions, policymakers can create a supportive environment that empowers all learners to thrive and succeed.

As European educational policy continues to evolve, it is essential to embrace a holistic perspective that prioritizes inclusivity, equity, and social justice. By doing so, Europe can pave the way for a more diverse and empowered generation of learners, breaking down barriers and shaping a brighter future for education.

**Funding:** This research is associated with the project funded by EDU2014-52702-R Project (Ministry of economy and competitivity of Spain) and the Author Doctoral Thesis. But funding has being fully assumed by the authors. The APC was funded by National University of Distance Education IOAP program, and partially paid by author.

**Conflicts of Interest:** The authors declare no conflict of interest.

## Notes

[1]    Tranlated from Spanish document
[2]    Tranlated from Spanish document
[3]    Tranlated from Spanish document

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
