# Peer review of "The European Educational Policy and Early School Leaving: A Conceptual Analysis from the Perspective of the Leaving Subject"

_philosophies, doi:10.3390/philosophies8050086_

Round 1

Reviewer 1 Report

Excellent presentation and critical analysis of the history and current policy regarding ESL--clearly and thoughtfully articulated and defended.  One point to consider for revision:  the authors mentioned equity and social justice as important considerations in the analysis of these policies, however, they do not provide a theory of social justice that would frame the analysis.  The prevention of such a theoretical framework for analysis in terms of social justice would strengthened the article. 

Author Response

Dear Reviewer,

I sincerely appreciate your valuable time and dedication in reviewing this manuscript. All of your insightful revisions have been thoughtfully considered and have contributed significantly to enhancing its overall quality. New additions have been highlighted in green, while modifications are marked in yellow.

Best regards,

The Author

Reviewer 2 Report

The article is a very interesting approach to the issue of early school leaving and presents an overview of different philosophical perspectives of perceiving the subject who leaves school prematurely and the concept of lifelong learning, encouraging a deeper reflection on the presented problem and social justice.

However, even though the article contains references to feminist and Marxist perspectives and mentions certain groups to which these perspectives refer, it seems quite general and exclusionary, not mentioning certain groups which often leave school prematurely and face barriers to participating in lifelong learning, such as ethnic minorities or persons with disabilities.

The title of text is consistent with the issue presented.

The manuscript would benefit from including more recent references. Why does the policy analysis stop at ET2020 objectives? There are now new objectives for the 2021-2030 period. Including some reference to the new strategic education framework would make the text more up-to-date. (e.g. Council Resolution on a strategic framework for European cooperation in education and training towards the European Education Area and beyond (2021-2030) 2021/C 66/01 (OJ C, C/66, 26.02.2021, p. 1, CELEX: https://eur-lex.europa.eu/legal-content/EN/TXT/?uri=CELEX:32021G0226(01))

The article is well written and mostly easy to understand, but certain fragments need clarifying or rephrasing (this is highlighted and in the attached document, with comments):
- the Great Depression after the Second World War (p. 3)
- Other European theorists (p. 3)
- Lisbon Council (p. 4-5)
- young people who leave school are classified as "at risk" (p. 4)
leaving students (p.12)

Quotations used in the text (pages 2, 5, 6) need revision, as the wording used by the Author(s) is not the wording used in the original cited texts.

Author Response

(The authors gave the same response as above.)
